# Overlearning Reveals Sensitive Attributes

**Congzheng Song**
Cornell University
cs2296@cornell.edu

**Vitaly Shmatikov**
Cornell Tech
shmat@cs.cornell.edu

## Abstract

"Overlearning" means that a model trained for a seemingly simple objective implicitly learns to recognize attributes and concepts that are (1) not part of the learning objective, and (2) sensitive from a privacy or bias perspective. For example, a binary gender classifier of facial images also learns to recognize races—even races that are not represented in the training data—and identities.

We demonstrate overlearning in several vision and NLP models and analyze its harmful consequences. First, inference-time representations of an overlearned model reveal sensitive attributes of the input, breaking privacy protections such as model partitioning. Second, an overlearned model can be "re-purposed" for a different, privacy-violating task even in the absence of the original training data.

We show that overlearning is intrinsic for some tasks and cannot be prevented by censoring unwanted attributes. Finally, we investigate where, when, and why overlearning happens during model training.

## 1 Introduction

We demonstrate that representations learned by deep models when training for seemingly simple objectives reveal privacy- and bias-sensitive attributes that are not part of the specified objective. These unintentionally learned concepts are neither finer-, nor coarse-grained versions of the model's labels, nor statistically correlated with them. We call this phenomenon **overlearning**. For example, a binary classifier trained to determine the gender of a facial image also learns to recognize races (including races not represented in the training data) and even identities of individuals.

Overlearning has two distinct consequences. First, the model's inference-time representation of an input reveals the input's sensitive attributes. For example, a facial recognition model's representation of an image reveals if two specific individuals appear together in it. Overlearning thus breaks inference-time privacy protections based on model partitioning (Osia et al., 2018; Chi et al., 2018; Wang et al., 2018). Second, we develop a new, transfer learning-based technique to "re-purpose" a model trained for benign task into a model for a different, privacy-violating task. This shows the inadequacy of privacy regulations that rely on explicit enumeration of learned attributes.

Overlearning is intrinsic for some tasks, i.e., it is not possible to prevent a model from learning sensitive attributes. We show that if these attributes are censored (Xie et al., 2017; Moyer et al., 2018), the censored models either fail to learn their specified tasks, or still leak sensitive information. We develop a new de-censoring technique to extract information from censored representations. We also show that overlearned representations enable recognition of sensitive attributes that are not present in the training data. Such attributes cannot be censored using any known technique. This shows the the inadequacy of censoring as a privacy protection technology.

To analyze where and why overlearning happens, we empirically show how general features emerge in the lower layers of models trained for simple objectives and conjecture an explanation based on the complexity of the training data.

## 2 Background

We focus on supervised deep learning. Given an input $x$, a model $M$ is trained to predict the target $y$ using a discriminative approach. We represent the model $M = C \circ E$ as a feature extractor

(encoder) $E$ and classifier $C$. The representation $z = E(x)$ is passed to $C$ to produce the prediction by modeling $p(y|z) = C(z)$. Since $E$ can have multiple layers of representation, we use $E_l(x) = z_l$ to denote the model's internal representation at layer $l$; $z$ is the representation at the last layer.

***Model partitioning*** splits the model into a local, on-device part and a remote, cloud-based part to improve scalability of inference (Lane & Georgiev, 2015; Kang et al., 2017) and protect privacy of inputs into the model (Li et al., 2017; Osia et al., 2018; Chi et al., 2018; Wang et al., 2018). For privacy, the local part of the model computes a representation, censors it as described below, and sends it to the cloud part, which computes the model's output.

***Censoring representations.*** The goal is to encode input $x$ into a representation $z$ that does not reveal unwanted properties of $x$, yet is expressive enough to predict the task label $y$. Censoring has been used to achieve transform-invariant representations for computer vision, bias-free representations for fair machine learning, and privacy-preserving representations that hide sensitive attributes.

A straightforward censoring approach is based on *adversarial training* (Goodfellow et al., 2014). It involves a mini-max game between a discriminator $D$ trying to infer $s$ from $z$ during training and an encoder and classifier trying to infer the task label $y$ while minimizing the discriminator's success (Edwards & Storkey, 2016; Iwasawa et al., 2016; Hamm, 2017; Xie et al., 2017; Li et al., 2018; Coavoux et al., 2018; Elazar & Goldberg, 2018). The game is formulated as:

$$\min_{E,C} \max_D \mathbb{E}_{x,y,s}[\gamma \log p(s|z = E(x)) - \log p(y|z = E(x))] \tag{1}$$

where $\gamma$ balances the two log likelihood terms. The inner optimization maximizes $\log p(s|z = E(x))$, i.e., the discriminator's prediction of the sensitive attribute $s$ given a representation $z$. The outer optimization, on the other hand, trains the encoder and classifier to minimize the log likelihood of the discriminator predicting $s$ and maximize that of predicting the task label $y$.

Another approach casts censoring as a single *information-theoretical* objective. The requirement that $z$ not reveal $s$ can be formalized as an independence constraint $z \perp s$, but independence is intractable to measure in practice, thus the requirement is relaxed to a constraint on the mutual information between $z$ and $s$ (Osia et al., 2018; Moyer et al., 2018). The overall training objective of censoring $s$ and predicting $y$ from $z$ is formulated as:

$$\max I(z,y) - \beta I(z,x) - \lambda I(z,s) \tag{2}$$

where $I$ is mutual information and $\beta, \lambda$ are the balancing coefficients; $\beta = 0$ in (Osia et al., 2018). The first two terms $I(z,y) - \beta I(z,x)$ is the objective of variational information bottleneck (Alemi et al., 2017), the third term is the relaxed independence constraint of $z$ and $s$.

Intuitively, this objective aims to maximize the information of $y$ in $z$ as per $I(z,y)$, forget the information of $x$ in $z$ as per $-\beta I(z,x)$, and remove the information of $s$ in $z$ as per $-\lambda I(z,s)$. This objective has an analytical lower bound (Moyer et al., 2018):

$$\mathbb{E}_{x,s}[\mathbb{E}_{z,y}[\log p(y|z)] - (\beta + \lambda)KL[q(z|x)||q(z)] - \lambda \mathbb{E}_z[\log p(x|z,s)]] \tag{3}$$

where $KL$ is Kullback-Leibler divergence and $\log p(x|z,s)$ is the reconstruction likelihood of $x$ given $z$ and $s$. The conditional distributions $p(y|z) = C(z)$, $q(z|x) = E(x)$ are modeled as in adversarial training and $p(x|z,s)$ is modeled with a decoder $R(z,s) = p(x|z,s)$.

All known censoring techniques require a "blacklist" of attributes to censor, and inputs with these attributes must be represented in the training data. Censoring for fairness is applied to the model's final layer to make its output independent of the sensitive attributes or satisfy a specific fairness constraint (Zemel et al., 2013; Louizos et al., 2016; Madras et al., 2018; Song et al., 2019). In this paper, we use censoring not for fairness but to demonstrate that models cannot be prevented from learning to recognize sensitive attributes. To show this, we apply censoring to different layers, not just the output.

## 3 EXPLOITING OVERLEARNING

We demonstrate two different ways to exploit overlearning in a trained model $M$. The inference-time attack (Section 3.1) applies $M$ to an input and uses $M$'s representation of that input to predict its sensitive attributes. The model-repurposing attack (Section 3.2) uses $M$ to create another model that, when applied to an input, directly predicts its sensitive attributes.

---

**Inferring $s$ from representation:**
1: **Input:** Adversary's auxiliary dataset $\mathcal{D}_{\text{aux}}$, black-box oracle $E$, observed $z^\star$
2: $\mathcal{D}_{\text{attack}} \leftarrow \{(E(x), s) \mid (x, s) \in \mathcal{D}_{\text{aux}}\}$
3: Train attack model $M_{\text{attack}}$ on $\mathcal{D}_{\text{attack}}$
4: **return** prediction $\hat{s} = M_{\text{attack}}(z^\star)$

**Adversarial re-purposing:**
1: **Input:** Model $M$ for the original task, transfer dataset $\mathcal{D}_{\text{transfer}}$ for the new task
2: Build $M_{\text{transfer}} = C_{\text{transfer}} \circ E_l$ on layer $l$
3: Fine-tune $M_{\text{transfer}}$ on $\mathcal{D}_{\text{transfer}}$
4: **return** transfer model $M_{\text{transfer}}$

Figure 1: Pseudo-code for inference from representation and adversarial re-purposing

---

**Algorithm 1** De-censoring representations

---

1: **Input:** Auxiliary dataset $\mathcal{D}_{\text{aux}}$, black-box oracle $E$, observed representation $z^\star$
2: Train auxiliary model $M_{\text{aux}} = E_{\text{aux}} \circ C_{\text{aux}}$ on $\mathcal{D}_{\text{aux}}$
3: Initialize transform model $T$, inference attack model $M_{\text{attack}}$
4: **for** each training iteration **do**
5:      Sample a batch of data $(x, s)$ from $\mathcal{D}_{\text{aux}}$ and compute $z = E(x), z_{\text{aux}} = E_{\text{aux}}(x)$
6:      Update $T$ on the batch of $(z, z_{\text{aux}})$ with loss $||T(z) - z_{\text{aux}}||_2^2$
7:      Update $M_{\text{attack}}$ on the batch of $(T(z), s)$ with cross-entropy loss
8: **end for**
9: **return** prediction $\hat{s} = M_{\text{attack}}(T(z^\star))$

---

## 3.1 Inferring sensitive attributes from representation

We measure the leakage of sensitive properties from the representations of overlearned models via the following attack. Suppose an adversary can observe the representation $z^\star$ of a trained model $M$ on input $x^\star$ at inference time but cannot observe $x^\star$ directly. This scenario arises in practice when model evaluation is partitioned in order to protect privacy of inputs—see Section 2. The adversary wants to infer some property $s$ of $x^\star$ that is not part of the task label $y$.

We assume that the adversary has an auxiliary set $\mathcal{D}_{\text{aux}}$ of labeled $(x, s)$ pairs and black-box oracle $E$ to compute the corresponding $E(x)$. The purpose of $\mathcal{D}_{\text{aux}}$ is to help the adversary recognize the property of interest in the model's representations; it need not be drawn from the same dataset as $x^\star$. The adversary uses supervised learning on the $(E(x), s)$ pairs to train an attack model $M_{\text{attack}}$. At inference time, the adversary predicts $\hat{s}$ from the observed $z^\star$ as $M_{\text{attack}}(z^\star)$.

***De-censoring.*** If the representation $z$ is "censored" (see Section 2) to reduce the amount of information it reveals about $s$, the direct inference attack may not succeed. We develop a new, learning-based *de-censoring* approach (see Algorithm 1) to convert censored representations into a different form that leaks more information about the property of interest. The adversary trains $M_{\text{aux}}$ on $\mathcal{D}_{\text{aux}}$ to predict $s$ from $x$, then transforms $z$ into the input features of $M_{\text{aux}}$.

We treat de-censoring as an optimization problem with a feature space $L_2$ loss $||T(z) - z_{\text{aux}}||_2^2$, where $T$ is the transformer that the adversary wants to learn and $z_{\text{aux}}$ is the uncensored representation from $M_{\text{aux}}$. Training with a feature-space loss has been proposed for synthesizing more natural images by matching them with real images (Dosovitskiy & Brox, 2016; Nguyen et al., 2016). In our case, we match censored and uncensored representations. The adversary can then use $T(z)$ as an uncensored approximation of $z$ to train an inference model $M_{\text{attack}}$ and infer property $s$ as $M_{\text{attack}}(T(z^\star))$.

## 3.2 Re-purposing models to predict sensitive attributes

To re-purpose a model—for example, to convert a model trained for a benign task into a model that predicts a sensitive attribute—we can use features $z_l$ in any layer of $M$ as the feature extractor and connect a new classifier $C_{\text{transfer}}$ to $E_l$. The transferred model $M_{\text{transfer}} = C_{\text{transfer}} \circ E_l$ is fine-tuned on another, small dataset $\mathcal{D}_{\text{transfer}}$, which in itself is not sufficient to train an accurate model for the new task. Utilizing features learned by $M$ on the original $\mathcal{D}$, $M_{\text{transfer}}$ can achieve better results than models trained from scratch on $\mathcal{D}_{\text{transfer}}$.

Feasibility of model re-purposing complicates the application of policies and regulations such as GDPR (EU, 2018). GDPR requires data processors to disclose every purpose of data collection and obtain consent from the users whose data was collected. We show that, given a trained model, it is not possible to determine—nor, consequently, disclose or obtain user consent for—what the model

Table 1: Summary of datasets and tasks. Cramer's V captures statistical correlation between $y$ and $s$ (0 indicates no correlation and 1 indicates perfectly correlated).

| Dataset | Health | UTKFace | FaceScrub | Places365 | Twitter | Yelp | PIPA |
|---|---|---|---|---|---|---|---|
| Target $y$ | CCI | gender | gender | in/outdoor | age | review score | facial IDs |
| Attribute $s$ | age | race | facial IDs | scene type | author | author | IDs together |
| Cramer's V | 0.149 | 0.035 | 0.044 | 0.052 | 0.134 | 0.033 | n/a |

has learned. Learning per se thus cannot be a regulated "purpose" of data collection. Regulators must be aware that even if the original training data has been erased, a model can be re-purposed for a different objective, possibly not envisioned at the time of original data collection. We discuss this further in Section 6.

# 4 EXPERIMENTAL RESULTS

## 4.1 DATASETS, TASKS, AND MODELS

**Health** is the Heritage Health dataset (Heritage Health Prize) with medical records of over 55,000 patients, binarized into 112 features with age information removed. The task is to predict if Charlson Index (an estimate of patient mortality) is greater than zero; the sensitive attribute is age (binned into 9 ranges).

**UTKFace** is a set of over 23,000 face images labeled with age, gender, and race (UTKFace; Zhang et al., 2017). We rescaled them into $50{\times}50$ RGB pixels. The task is to predict gender; the sensitive attribute is race.

**FaceScrub** is a set of face images labeled with gender (FaceScrub). Some URLs are expired, but we were able to download 74,000 images for 500 individuals and rescale them into $50{\times}50$ RGB pixels. The task is to predict gender; the sensitive attribute is identity.

**Places365** is a set of 1.8 million images labeled with 365 fine-grained scene categories. We use a subset of 73,000 images, 200 per category. The task is to predict whether the scene is indoor or outdoor; the sensitive attribute is the fine-grained scene label.

**Twitter** is a set of tweets from the PAN16 dataset (Rangel et al., 2016) labeled with user information. We removed tweets with fewer than 20 tokens and users with fewer than 50 tweets, yielding a dataset of over 46,000 tweets from 151 users with an over 80,000-word vocabulary. The task is to predict the age of the user given a tweet; the sensitive attribute is the author's identity.

**Yelp** is a set of Yelp reviews labeled with user identities (Yelp Open Dataset). We removed users with fewer than 1,000 reviews and reviews with more than 200 tokens, yielding a dataset of over 39,000 reviews from 137 users with an over 69,000-word vocabulary. The task is to predict the review score between 1 to 5; the sensitive attribute is the author's identity.

**PIPA** is a set of over 60,000 photos of 2,000 individuals gathered from public Flickr photo albums (Piper project page; Zhang et al., 2015). Each image can include one or more individuals. We cropped their head regions using the bounding boxes in the image annotations. The task is to predict the identity given the head region; the sensitive attribute is whether two head regions are from the same photo.

**Models.** For Health, we use a two-layer fully connected (FC) neural network with 128 and 32 hidden units, respectively, following (Xie et al., 2017; Moyer et al., 2018). For UTKFace and FaceScrub, we use a LeNet (LeCun et al., 1998) variant: three $3{\times}3$ convolutional and $2{\times}2$ max-pooling layers with 16, 32, and 64 filters, followed by two FC layers with 128 and 64 hidden units. For Twitter and Yelp, we use text CNN (Kim, 2014). For Places365 and PIPA, we use AlexNet (Krizhevsky et al., 2012) with convolutional layers pre-trained on ImageNet (Deng et al., 2009) and further add a $3{\times}3$ convolutional layer with 128 filters and $2{\times}2$ max-pooling followed by two FC layers with 128 and 64 hidden units, respectively.

Table 2: Accuracy of inference from representations (last FC layer). RAND is random guessing based on majority class labels; BASE is inference from the uncensored representation; ADV from the representation censored with adversarial training; IT from the information-theoretically censored representation.

| Dataset | Acc of predicting target $y$ | | | | Acc of inferring sensitive attribute $s$ | | | |
|---|---|---|---|---|---|---|---|---|
| | RAND | BASE | ADV | IT | RAND | BASE | ADV | IT |
| Health | 66.31 | 84.33 | 80.16 | 82.63 | 16.00 | 32.52 | 32.00 | 26.60 |
| UTKFace | 52.27 | 90.38 | 90.15 | 88.15 | 42.52 | 62.18 | 53.28 | 53.30 |
| FaceScrub | 53.53 | 98.77 | 97.90 | 97.66 | 1.42 | 33.65 | 30.23 | 10.61 |
| Places365 | 56.16 | 91.41 | 90.84 | 89.82 | 1.37 | 31.03 | 12.56 | 2.29 |
| Twitter | 45.17 | 76.22 | 57.97 | n/a | 6.93 | 38.46 | 34.27 | n/a |
| Yelp | 42.56 | 57.81 | 56.79 | n/a | 15.88 | 33.09 | 27.32 | n/a |
| PIPA | 7.67 | 77.34 | 52.02 | 29.64 | 68.50 | 87.95 | 69.96 | 82.02 |

## 4.2 INFERRING SENSITIVE ATTRIBUTES FROM REPRESENTATIONS

***Setup.*** We use 80% of the data for training the target models and 20% for evaluation. The size of the adversary's auxiliary dataset is 50% of the training data. Success of the inference attack is measured on the final FC layer's representation of test data. The baseline is inference from the uncensored representation. We also measure the success of inference against representations censored with $\gamma = 1.0$ for adversarial training and $\beta = 0.01, \lambda = 0.0001$ for information-theoretical censoring, following (Xie et al., 2017; Moyer et al., 2018).

For censoring with adversarial training, we simulate the adversary with a two-layer FC neural network with 256 and 128 hidden units. The number of epochs is 50 for censoring with adversarial training, 30 for the other models. We use the Adam optimizer with the learning rate of 0.001 and batch size of 128. For information-theoretical censoring, the model is based on VAE (Kingma & Welling, 2013; Moyer et al., 2018). The encoder $q(z|x)$ has the same architecture as the CNN models with all convolutional layers. On top of that, the encoder outputs a mean vector and a standard deviation vector to model the random variable $z$ with the re-parameterization trick. The decoder $p(x|z)$ has three de-convolution layers with up-sampling to map $z$ back to the same shape as the input $x$.

For our inference model, we use the same architecture as the censoring adversary. For the PIPA inference model, which takes two representations of faces and outputs a binary prediction of whether these faces appear in the same photo, we use two FC layers followed by a bilinear model: $p(s|z_1, z_2) = \sigma(h(z_1)Wh(z_2)^\top)$, where $z_1, z_2$ are the two input representations, $h$ is the two FC layers, and $\sigma$ is the sigmoid function. We train the inference model for 50 epochs with the Adam optimizer, learning rate of 0.001, and batch size of 128.

***Results.*** Table 2 reports the results. When representations are not censored, accuracy of inference from the last-layer representations is much higher than random guessing for all tasks, which means **models overlearn even in the higher, task-specific layers**. When representations are censored with adversarial training, accuracy drops for both the main and inference tasks. Accuracy of inference is much higher than in (Xie et al., 2017). The latter uses logistic regression, which is weaker than the training-time censoring-adversary network, whereas we use the same architecture for both the training-time and post-hoc adversaries. Information-theoretical censoring reduces accuracy of inference, but also damages main-task accuracy more than adversarial training for almost all models.

**Overlearning can cause a model to recognize even the sensitive attributes that are not represented in the training dataset**. Such attributes cannot be censored using any known technique. We trained a UTKFace gender classifier on datasets where all faces are of the same race. We then applied this model to test images with four races (White, Black, Asian, Indian) and attempted to infer the race attribute from the model's representations. Inference accuracy is 61.95%, 61.99%, 60.85% and 60.81% for models trained only on, respectively, White, Black, Asian, and Indian images—almost as good as the 62.18% baseline and much higher than random guessing (42.52%).

***Effect of censoring strength.*** Fig. 2 shows that stronger censoring does not help. On FaceScrub and Twitter with adversarial training, increasing $\gamma$ damages the model's accuracy on the main task, while

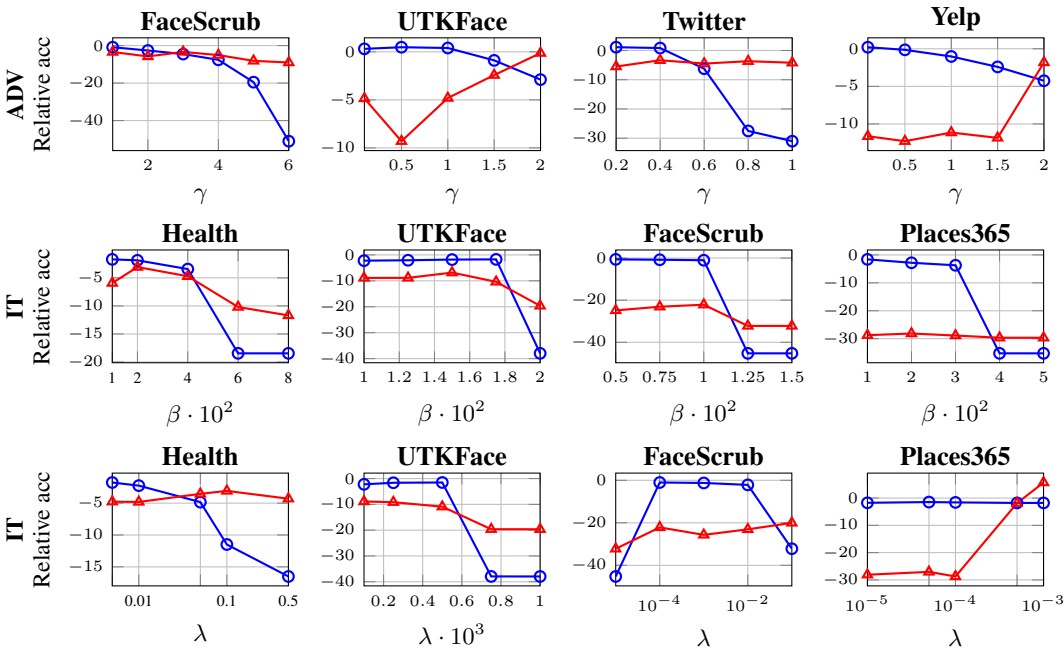

Figure 2: Reduction in accuracy due to censoring. Blue lines are the main task, red lines are the inference of sensitive attributes. First row is adversarial training with different $\gamma$ values; second and third row is information-theoretical censoring with different $\beta$ and $\lambda$ values respectively.

Table 3: Improving inference accuracy with de-censoring. $\delta$ is the increase from Table 2.

| Dataset | Health | UTKFace | FaceScrub | Places365 | Twitter | Yelp |
|---|---|---|---|---|---|---|
| ADV $+\delta$ | 32.55 +0.55 | 59.38 +6.10 | 40.37 +12.24 | 19.71 +7.15 | 36.55 +2.22 | 31.36 +4.04 |
| IT $+\delta$ | 27.05 +0.45 | 54.31 +1.01 | 16.40   +5.79 | 3.10 +0.81 | n/a | n/a |

accuracy of inference decreases slightly or remains the same. For UTKFace and Yelp, increasing $\gamma$ *improves* accuracy of inference. This may indicate that the simulated "adversary" during adversarial training overpowers the optimization process and censoring defeats itself.

For all models with information-theoretical censoring, increasing $\beta$ reduces the accuracy of inference but can lead to the model not converging on its main task. Increasing $\lambda$ results in the model not converging on the main task, without affecting the accuracy of inference, on Health, UTKFace and FaceScrub. This seems to contradict the censoring objective, but the reconstruction loss in Equation 2 dominates the other loss terms, which leads to poor divergence between conditional $q(z|x)$ and $q(z)$, i.e., information about $x$ is still retained in $z$.

***De-censoring.*** As described in Section 3.1, we developed a new technique to transform censored representations to make inference easier. We first train an auxiliary model on $\mathcal{D}_{\text{aux}}$ to predict the sensitive attribute from representations, using the same architecture as in the baseline models. The resulting uncensored representations from the last convolutional layer are the target for the de-censoring transformations. We use a single-layer fully connected neural network as the transformer and set the number of hidden units to the dimension of the uncensored representation. The inference model operates on top of the transformer network, with the same hyper-parameters as before.

Table 3 shows that de-censoring significantly boosts the accuracy of inference from representations censored with adversarial training. The boost is smaller against information-theoretical censoring because its objective not only censors $z$ with $I(z, s)$, but also forgets $x$ with $I(x, z)$. On the Health task, there is not much difference since the baseline attack is already similar to the attack on censored representations, leaving little room for improvement.

Table 4: Adversarial re-purposing. The values are differences between the accuracy of predicting sensitive attributes using a re-purposed model vs. a model trained from scratch.

| $|\mathcal{D}_{\text{transfer}}|/|\mathcal{D}|$ | Health | UTKFace | FaceScrub | Places365 | Twitter | Yelp | PIPA |
|---|---|---|---|---|---|---|---|
| 0.02 | -0.57 | 4.72 | 7.01 | 4.42 | 12.99 | 5.57 | 1.33 |
| 0.04 | 0.22 | 2.70 | 15.07 | 2.14 | 10.87 | 3.60 | 2.41 |
| 0.06 | -1.21 | 2.83 | 7.02 | 2.06 | 10.51 | 8.45 | 6.50 |
| 0.08 | -0.99 | 0.25 | 11.80 | 3.39 | 9.57 | 0.33 | 4.93 |
| 0.10 | 0.35 | 2.24 | 9.43 | 2.86 | 7.30 | 2.1 | 5.89 |

Table 5: The effect of censoring on adversarial re-purposing for FaceScrub with $\gamma = 0.5, 0.75, 1.0$. $\delta_A$ is the difference in the original-task accuracy (second column) between uncensored and censored models; $\delta_B$ is the difference in the accuracy of inferring the sensitive attribute (columns 3 to 7) between the models re-purposed from different layers and the model trained from scratch. Negative values mean reduced accuracy. Heatmaps on the right are linear CKA similarities between censored and uncensored representations. Numbers 0 through 4 represent layers conv1, conv2, conv3, fc4, and fc5. For each model censored at layer $i$ (x-axis), we measure similarity between the censored and uncensored models at layer $j$ (y-axis).

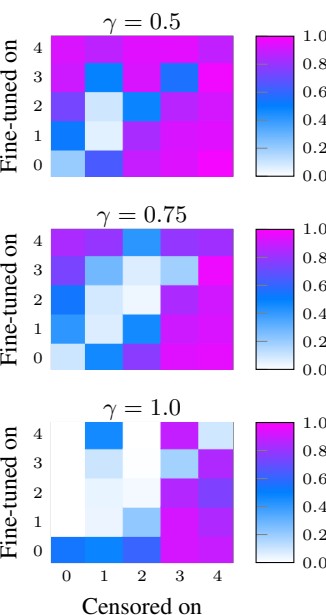

| Censored on $\gamma = 0.5$ | $\delta_A$ | $\delta_B$ when transferred from | | | | |
|---|---|---|---|---|---|---|
| | | conv1 | conv2 | conv3 | fc4 | fc5 |
| conv1 | -1.66 | -6.42 | -4.09 | -1.65 | 0.46 | -3.87 |
| conv2 | -2.87 | 0.95 | -1.77 | -2.88 | -1.53 | -2.22 |
| conv3 | -0.64 | 1.49 | 1.49 | 0.67 | -0.48 | -1.38 |
| fc4 | -0.16 | 2.03 | 5.16 | 6.73 | 6.12 | 0.54 |
| fc5 | 0.05 | 1.52 | 4.53 | 7.42 | 6.14 | 4.53 |
| $\gamma = 0.75$ | | | | | | |
| conv1 | -4.48 | -7.33 | -5.01 | -1.51 | -7.99 | -7.82 |
| conv2 | -6.02 | 0.44 | -7.04 | -5.46 | -5.94 | -5.82 |
| conv3 | -1.90 | 1.32 | 1.37 | 1.88 | 0.74 | -0.67 |
| fc4 | 0.01 | 3.65 | 4.56 | 5.11 | 4.44 | 0.91 |
| fc5 | -0.74 | 1.54 | 3.61 | 6.75 | 7.18 | 4.99 |
| $\gamma = 1$ | | | | | | |
| conv1 | -45.25 | -7.36 | -3.93 | -2.75 | -4.37 | -2.91 |
| conv2 | -20.30 | -3.28 | -5.27 | -7.03 | -6.38 | -5.54 |
| conv3 | -45.20 | -2.13 | -3.06 | -4.48 | -4.05 | -5.18 |
| fc4 | -0.52 | 1.73 | 5.19 | 4.80 | 5.83 | 1.84 |
| fc5 | -0.86 | 1.56 | 3.55 | 5.59 | 5.14 | 1.97 |

In summary, these results demonstrate that information about sensitive attributes unintentionally captured by the overlearned representations cannot be suppressed by censoring.

### 4.3 RE-PURPOSING MODELS TO PREDICT SENSITIVE ATTRIBUTES

To demonstrate that **overlearned representations can be picked up by a small set of unseen data to create a model for predicting sensitive attributes**, we re-purpose uncensored baseline models from Section 4.2 by fine-tuning them on a small $(2 - 10\%$ of $\mathcal{D})$ set $\mathcal{D}_{\text{transfer}}$ and compare with the models trained from scratch on $\mathcal{D}_{\text{transfer}}$. We fine-tune all models for 50 epochs with batch size of 32; the other hyper-parameters are as in Section 4.2. For all CNN models, we use the trained convolutional layers as the feature extractor and randomly initialize the other layers. Table 4 shows that the re-purposed models always outperform those trained from scratch. FaceScrub and Twitter exhibit the biggest gain.

***Effect of censoring.*** Previous work only censored the highest layer of the models. Model re-purposing can use any layer of the model for transfer learning. Therefore, to prevent re-purposing, inner layers must be censored, too. We perform the first study of inner-layers censoring and measure

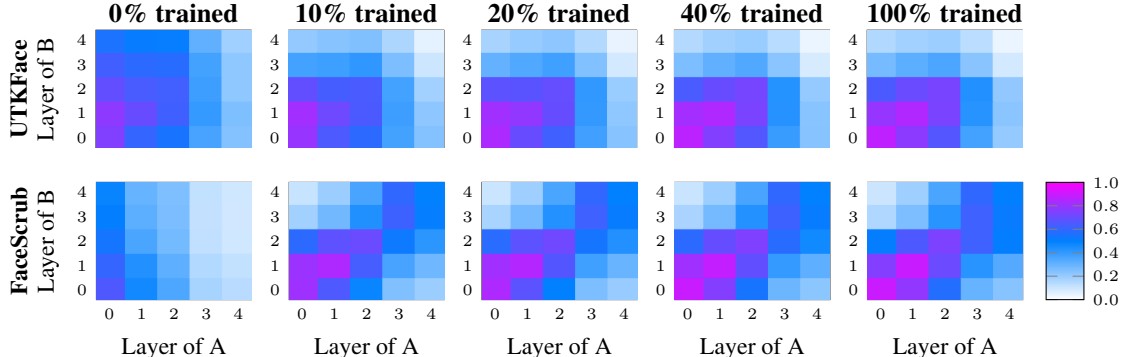

Figure 3: Pairwise similarities of layer representations between models for the original task (A) and for predicting a sensitive attribute (B). Numbers 0 through 4 denote layers conv1, conv2, conv3, fc4 and fc5.

its effect on both the original and re-purposed tasks. We use FaceScrub for this experiment and apply adversarial training to every layer with different strengths ($\gamma = 0.5, 0.75, 1.0$).

Table 5 summarizes the results. Censoring lower layers (conv1 to conv3) blocks adversarial re-purposing, at the cost of reducing the model's accuracy on its original task. Hyper-parameters must be tuned carefully, e.g. when $\gamma = 1$, there is a huge drop in the original-task accuracy.

To further investigate how censoring in one layer affects the representations learned across all layers, we measure per-layer similarity between censored and uncensored models using CKA, linear centered kernel alignment (Kornblith et al., 2019)—see Table 5. When censoring is applied to a specific layer, similarity for that layer is the smallest (values on the diagonal). When censoring lower layers with moderate strength ($\gamma = 0.5$ or $0.75$), similarity between higher layers is still strong; when censoring higher layers, similarity between lower layers is strong. Therefore, censoring can block adversarial re-purposing from a specific layer, but the adversary can still re-purpose representations in the other layer(s) to obtain an accurate model for predicting sensitive attributes.

## 4.4 WHEN, WHERE, AND WHY OVERLEARNING HAPPENS

To investigate when (during training) and where (in which layer) the models overlearn, we use linear CKA similarity (Kornblith et al., 2019) to compare the representations at different epochs of training between models trained for the original task (A) and models trained to predict a sensitive attribute (B). We use UTKFace and FaceScrub for these experiments.

Fig. 3 shows that lower layers of models A and B learn very similar features. This was observed in (Kornblith et al., 2019) for CIFAR-10 and CIFAR-100 models, but those tasks are closely related. In our case, the tasks are entirely different and B reveals the sensitive attribute while A does not. The similar low-level features are learned very early during training. There is little similarity between the low-level features of A and high-level features of B (and vice versa), matching intuition. Interestingly, on FaceScrub even the high-level features are similar between A and B.

We conjecture that one of the reasons for overlearning is structural complexity of the data. Previous work theoretically showed that over-parameterized neural networks favor simple solutions on structured data when optimized with SGD, where structure is quantified as the number of distributions (e.g., images from different identities) within each class in the target task (Li & Liang, 2018), i.e., the fewer distributions, the more structured the data. For data generated from more complicated distributions, networks learn more complex solutions, leading to the emergence of features that are much more general than the learning objective and, consequently, overlearning.

Fig. 4 shows that the representations of a gender classifier trained on the faces from 50 individuals are closer to the random initialization than the representations trained on the faces from 500 individuals (the hyper-parameters and the total number of training examples are the same in both cases). More complex training data thus results in more complex representations for the same objective.

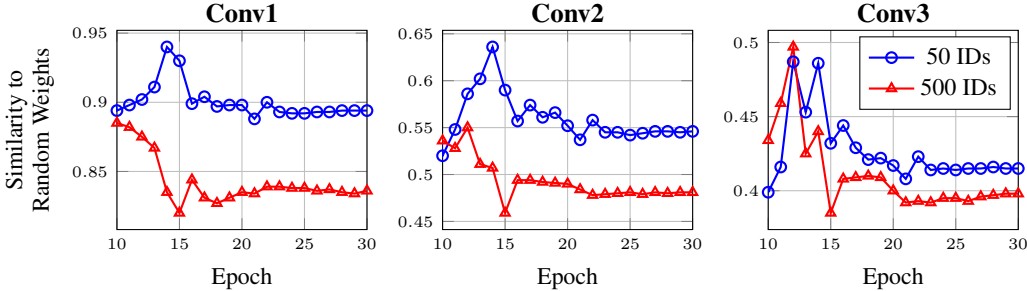

Figure 4: Similarity of layer representations of a partially trained gender classifier to a randomly initialized model before training. Models are trained on FaceScrub using 50 IDs (blue line) and 500 IDs (red line).

## 5 RELATED WORK

Prior work studied transferability of representations only between closely related tasks. Transferability of features between ImageNet models decreases as the distance between the base and target tasks grows (Yosinski et al., 2014), and performance of tasks is correlated to their distance from the source task (Azizpour et al., 2015). CNN models trained to distinguish coarse classes also distinguish their subsets (Huh et al., 2016). By contrast, we show that models trained for simple tasks implicitly learn privacy-sensitive concepts unrelated to the labels of the original task. Other than an anecdotal mention in the acknowledgments paragraph of (Kim et al., 2017) that logit-layer activations leak non-label concepts, this phenomenon has never been described in the research literature.

Gradient updates revealed by participants in distributed learning leak information about individual training batches that is uncorrelated with the learning objective (Melis et al., 2019). We show that overlearning is a generic problem in (fully trained) models, helping explain these observations.

There is a large body of research on learning disentangled representations (Bengio et al., 2013; Locatello et al., 2019). The goal is to separate the underlying explanatory factors in the representation so that it contains all information about the input in an interpretable structure. State-of-the-art approaches use variational autoencoders (Kingma & Welling, 2013) and their variants to learn disentangled representations in an unsupervised fashion (Higgins et al., 2017; Kumar et al., 2018; Kim & Mnih, 2018; Chen et al., 2018). By contrast, overlearning means that representations learned during supervised training for one task implicitly and automatically enable another task—without disentangling the representation on purpose during training.

Work on censoring representations aims to suppress sensitive demographic attributes and identities in the model's output for fairness and privacy. Techniques include adversarial training (Edwards & Storkey, 2016), which has been applied to census and health records (Xie et al., 2017), text (Li et al., 2018; Coavoux et al., 2018; Elazar & Goldberg, 2018), images (Hamm, 2017) and sensor data of wearables (Iwasawa et al., 2016). An alternative approach is to minimize mutual information between the representation and the sensitive attribute (Moyer et al., 2018; Osia et al., 2018). Neither approach can prevent overlearning, except at the cost of destroying the model's accuracy. Furthermore, these techniques cannot censor attributes that are not represented in the training data. We show that overlearned models recognize such attributes, too.

## 6 CONCLUSIONS

We demonstrated that models trained for seemingly simple tasks implicitly learn concepts that are not represented in the objective function. In particular, they learn to recognize sensitive attributes, such as race and identity, that are statistically orthogonal to the objective. The failure of censoring to suppress these attributes and the similarity of learned representations across uncorrelated tasks suggest that overlearning may be intrinsic, i.e., learning for some objectives may not be possible without recognizing generic low-level features that enable other tasks, including inference of sensi-

tive attributes. For example, there may not exist a set of features that enables a model to accurately determine the gender of a face but not its race or identity.

This is a challenge for regulations such as GDPR that aim to control the purposes and uses of machine learning technologies. To protect privacy and ensure certain forms of fairness, users and regulators may desire that models not learn some features and attributes. If overlearning is intrinsic, it may not be technically possible to enumerate, let alone control, what models are learning. Therefore, regulators should focus on ensuring that models are applied in a way that respects privacy and fairness, while acknowledging that they may still recognize and use sensitive attributes.

***Acknowledgments.*** This research was supported in part by NSF grants 1611770, 1704296, and 1916717, the generosity of Eric and Wendy Schmidt by recommendation of the Schmidt Futures program, and a Google Faculty Research Award.

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
