# OpenReview forum: "Overlearning Reveals Sensitive Attributes"
_ICLR.cc/2020/Conference — Accept (Poster)_

### Official Review · AnonReviewer1 · 2019-10-22
**Official Blind Review #1**

**Rating:** 6

**Review:**

This paper demonstrates some limitations of censoring for privacy with respect to sensitive attributes. In particular, the authors show that censoring reduces, but does not eliminate, the ability of a neural network to infer private/sensitive attributes, e.g. to infer race from a model aiming to predict gender. Part of the proposed method is a component that performs de-censoring using an auxiliary dataset. The authors show that censoring strength often does reduce the ability to infer sensitive attributes, but also affects the ability to perform the main (non-sensitive) task; and in some cases, may actually increase ability to infer sensitive attributes. This type of work is important in that privacy is of growing importance, and so is the risk to privacy; this particular work is well carried out.

One concern: In Sec. 3.1, there is an assumption of the availability of D_{aux}. How realistic is this assumption?

**Experience Assessment:**

I do not know much about this area.

**Review Assessment: Checking Correctness Of Derivations And Theory:**

I assessed the sensibility of the derivations and theory.

**Review Assessment: Checking Correctness Of Experiments:**

I assessed the sensibility of the experiments.

**Review Assessment: Thoroughness In Paper Reading:**

I read the paper thoroughly.

---

> ### Author Response · Authors · 2019-11-14
> **Response to Reviewer #1**
>
> Thanks for the review!
> With the growing availability of public datasets, if the adversary knows only the input domain (e.g., face images), today it is relatively easy to find public data with the desired target attribute (e.g., race).

---

### Official Review · AnonReviewer2 · 2019-10-26
**Official Blind Review #2**

**Rating:** 1

**Review:**

This paper highlights the problem of model overlearning - learning more than it is trained to do. Thus, there is leak of privacy and sensitive attributes of images during test/ inference time.

Pros:
1. The paper is well written and easy to follow

Cons:
1. There is very little novelty in this paper - the notion of overlearning is well established in the literature (Osia et al., 2018; Chi et al., 2018; Wang et al., 2018). This paper merely reinstates, what is being already told in the literature.

2. In fact, there are many defence mechanisms proposed in the literature, for example "Anonymizing k Facial Attributes via Adversarial Perturbations" IJCAI 2018 - where the authors are performing data perturbations to minimize overlearning. This paper does not suggest or propose any method for solving the issue of overlearning

In summary, this paper repeats a well established problem of overlearning, showing experiments that are already shown in literature with known datasets, and also NOT proposing a solution to minimize overlearning (as many papers already proposed in literature).

3. Additionally, the experiments are very weak - the authors still perform experiments using LeNet variants and AlexNet, and for text using a textCNN. Why did the authors not perform experiments using more state-of the art CNN/RNN models. Did they not observe overlearning in these models?

4. As for section 4.4, it is pretty understood that lower layers of a DL model, learns very basic low-level features from the images such as edges, corner. Reinstating that, and calling it the reason for overlearning is not very convincing.

As of now, I find the paper very weak, till a solution to avoid overlearning is not proposed as a part of this paper.

**Experience Assessment:**

I have published in this field for several years.

**Review Assessment: Checking Correctness Of Derivations And Theory:**

I carefully checked the derivations and theory.

**Review Assessment: Checking Correctness Of Experiments:**

I assessed the sensibility of the experiments.

**Review Assessment: Thoroughness In Paper Reading:**

I read the paper thoroughly.

---

> ### Author Response · Authors · 2019-11-14
> **Response to Reviewer #2**
>
> Thanks for the review! We clarify several essential points, most of which are already covered in the paper:
>
> (1) Osia et al., Chi et al., Wang et al. do not describe overlearning, i.e., systematic leakage of multiple concepts orthogonal to the learning objective from the model’s intermediate layers.  They censor a known attribute in a single layer, but we explain in Section 2 that this cannot prevent overlearning because (a) the censor must know which attribute to censor, and (b) this attribute must be represented in the training data.  By contrast, Section 4.2 shows that overlearned representations enable recognition of attributes not present in the training data.  Censoring them is impossible with any existing method.
>
> (2. a) The adversarial perturbation paper [1] uses adversarial examples against a trained attribute classifier to prevent inferences by the same classifier (VGGFace). An inference-time defense cannot prevent overlearning (which is a training issue).  Further, to block leakage, the adversarial examples would have to transfer across all layers in all possible attribute classifiers, not just a specific classifier known to the defender as in [1]. There is no evidence in [1] that their adversarial examples defend against anything other than a single known classifier.
> (2. b) All known perturbation methods [2,3,4] work only on images or synthetic binary or real-valued data.  They cannot obfuscate discrete data such as texts and documents yet we show that models trained on such data overlearn and leak, too.  Further, a data holder with limited data for each attribute value (e.g., faces of one specific individual) cannot train an obfuscator that removes this sensitive attribute. If the obfuscator/privatizer is trained by a third party, then the obfuscator itself (typically a neural-network encoder) can leak information via adversarial re-purposing as we show in Section 4.3.
>
> (3) We chose model architectures that are adequate for the tasks in question, sufficient to demonstrate overlearning, and match prior work on censoring. To the best of our knowledge, the current implementations of censoring methods have been applied only to simpler models and tasks than ours [5,6,7].
>
> (4) We did not say that learning basic visual features in the lower layer of CNN is the “reason” for overlearning. We use them only to show that lower-layer representations can overlearn and be repurposed, too. Our conjectured reason for overlearning is that SGD finds more complex solutions for less structured data.
>
> (5) We do not propose a defense because we offer evidence that overlearning is an intrinsic problem and thus cannot be eliminated.  This also explains why Osia et al., Chi et al., Wang et al. do not solve the problem identified in our paper: they cannot prevent the leakage of all sensitive attributes in all layers without destroying the model’s accuracy on its given, non-sensitive task.
>
> Reference:
> [1] S. Chhabra et al. Anonymizing k Facial Attributes via Adversarial Perturbations. IJCAI, 2018
> [2] C. Huang et al. Context-aware generative adversarial privacy. Entropy, 2017
> [3] A. Tripathy et al. Privacy-preserving adversarial networks, 2018
> [4] M. Bertran et al. Adversarially learned representations for information obfuscation and inference. ICML, 2019.
> [5] Q. Xie et al. Controllable Invariance through Adversarial Feature Learning, NeurIPS, 2017
> [6] D. Moyer et al. Invariant Representations without Adversarial Training, NeurIPS, 2018
> [7] S.A. Osia et al. Deep Private-Feature Extraction, Transactions on Knowledge and Data Engineering, 2018

---

### Official Review · AnonReviewer3 · 2019-10-28
**Official Blind Review #3**

**Rating:** 6

**Review:**

This paper introduces the problem of overlearning, which can be thought of as unintended transfer learning from a (victim) source model to a target task that the source model’s creator had not intended its model to be used for. The paper raises good points about privacy legislation limitations due to the fact that overlearning makes it impossible to foresee future uses of a given dataset.

Background on censoring is well developed, and helps position the submission. However, the relation to transfer learning is not sufficiently outlined in the introduction. In some ways, overlearning is a form of unintended transfer learning. In particular, the connection appears explicit in the case of model repurposing (Section 3.2). Editing the introduction to tease apart this relationship to transfer learning would help readers forge an intuition for what the paper considers.

While the de-censoring algorithm is intuitive, it is not clear what assumptions are being made because of ambiguous notation in Algorithm 1. How does Algorithm 1 train E_aux and C_aux? Does that step assume the adversary has knowledge of the algorithm use to train E and C? Descriptions of the experimental setup from Section 4.2 seem to indicate that this is the case. This is not necessarily an issue if the proposed attack is demonstrating a limitation of learning rather than a practical attack.

Experiments overall show that censoring does not mitigate overlearning in a way that is robust to de-censoring. One aspect of the experiments that is unclear is the auxiliary dataset: what is the auxiliary dataset used by the adversary for each of the datasets considered in experiments?

Question: how does simultaneous censoring of all layers affect overlearning and repurposing?

Question: what is the connection between learning more complex representations and overlearning? Is the intuition that if the representation is more complex, it is more likely to contain features useful to identify the sensitive attribute?

The paper is overall well-written. Some parts of the paper omit important details (perhaps due to space constraints?), but an editorial pass should address most of these. Detailed feedback:

1 / Explaining what model partitioning means in this context would help make the introduction more self-contained.

1 / Is overlearning specific to attributes that raise fairness issues? Or is this phenomenon more general? The “Censoring representations” paragraph on page 2 seems to indicate that the phenomenon is more general.

5/ Is accuracy the best metric to report (e.g., for race attribute prediction) given that the random guessing figure suggests the data is not balanced across attribute values?

6/ Why is the effect not monotonic in Table 4? Were multiple runs averaged?

**Experience Assessment:**

I have published in this field for several years.

**Review Assessment: Checking Correctness Of Derivations And Theory:**

N/A

**Review Assessment: Checking Correctness Of Experiments:**

I carefully checked the experiments.

**Review Assessment: Thoroughness In Paper Reading:**

I read the paper thoroughly.

---

> ### Author Response · Authors · 2019-11-14
> **Response to Reviewer #3**
>
> Thanks for the review!
>
> (1)  $E_{aux}$ and $C_{aux}$ have similar architecture to $E$ and $C$ except for the output layer. They can have different architectures as long as the dimension for z is the same, in order to calculate the feature space $L_2$ loss.
>
> (2) Following [1,2], the auxiliary dataset in our experiment is a subset of the training data for the inference-time attack, and a subset of the held-out data for adversarial re-purposing.
>
> (3) Censoring all layers can mitigate overlearning but it prevents the model from learning at all: training fails to converge and the model does not learn its primary objective.
>
> (4) More complex representations lead to overlearning and leaking sensitive information.
>
> (5) Overlearning is a more general problem.  Representations that leak demographic information might be problematic from the fairness perspective, but fairness has a different set of goals (equal odds, equal opportunity, etc.). Whether censoring demographic attributes, i.e. “fairness through blindness,” can achieve these goals is still controversial in the fairness research community.
>
> (6) We report accuracy to be consistent with prior work on censoring [3,4]. In all tasks, the adversary’s accuracy of predicting the sensitive attribute (due to overlearning) is significantly higher than the baseline.
>
> (7) In total 5 runs were done.  In general, the absolute performance numbers for training from scratch and adversarial re-purposing increased monotonically with data size but not necessarily the gap (numbers in the table) between them.
>
> Reference:
> [1] https://github.com/dcmoyer/inv-rep
> [2] https://github.com/qizhex/Controllable-Invariance
> [3] Q. Xie et al. Controllable Invariance through Adversarial Feature Learning, NeurIPS, 2017
> [4] D. Moyer et al. Invariant Representations without Adversarial Training, NeurIPS, 2018

---

### Decision · Program_Chairs · 2019-12-19

**Decision:**

Accept (Poster)

**Comment:**

This paper introduces the problem of overlearning, which can be thought of as unintended transfer learning from a (victim) source model to a target task that the source model’s creator had not intended its model to be used for. The paper raises good points about privacy legislation limitations due to the fact that overlearning makes it impossible to foresee future uses of a given dataset.

Please incorporate the revisions suggested in the reviews to add clarity to the overlearning versus censoring confusion addressed by the reviewers.